# VC Theoretical Explanation of Double Descent

## Abstract

There has been growing interest in generalization performance of large multilayer neural networks that can be trained to achieve zero training error, while generalizing well on test data. This regime is known as 'second descent' and it appears to contradict conventional view that optimal model complexity should reflect optimal balance between underfitting and overfitting, aka the bias-variance trade-off. This paper presents VC-theoretical analysis of double descent and shows that it can be fully explained by classical VC generalization bounds. We illustrate an application of analytic VC-bounds for modeling double descent for classification problems, using empirical results for several learning methods, such as SVM, Least Squares, and Multilayer Perceptron classifiers. In addition, we discuss several possible reasons for misinterpretation of VC-theoretical results in the machine learning community.

## 1 Introduction

There have been many recent successful applications of Deep Learning (DL). However, at present time, various DL methods are driven mainly by heuristic improvements, while theoretical and conceptual understanding of this technology remains limited. For example, large neural networks can be trained to fit available data (achieving zero training error) and still achieve good generalization for test data. This contradicts conventional statistical wisdom that overfitting leads to poor generalization. This phenomenon has been systematically described by Belkin, et al. [1] who introduced the appropriate terminology ('double descent') and pointed out the difference between the classical regime (first descent) and the modern one (second descent). This disagreement between the classical statistical view and modern machine learning practice provides motivation for new theoretical explanations of the generalization ability of DL networks and other over-parameterized estimators. Several different explanations include: special properties of multilayer network parameterization [2], choosing proper inductive bias during second descent [1], effect of Stochastic Gradient Descent (SGD) training [3, 4, 5], the effect of various heuristics (used for training) on generalization [6], and the effect of margin on generalization [7]. The current consensus view on the 'generalization paradox' in DL networks is summarized below:

- Existing indices for model complexity (or capacity), such as VC-dimension, cannot explain generalization performance of DL networks.

- 'Classical' theories developed in ML and statistics cannot explain generalization performance of DL networks. In particular, classical VC generalization bounds cannot be used to explain double descent. Specifically, the ability of large DL networks to achieve zero training error (during second descent mode) effectively 'rules out all of the VC-dimension arguments as a possible explanation for the generalization performance of state-of-the-art neural networks' [3].

Submitted to 36th Conference on Neural Information Processing Systems (NeurIPS 2022). Do not distribute.

This paper demonstrates that these assertions are incorrect, and that classical VC-theoretical results can fully explain generalization performance of DL networks, including 'double descent', for classification problems. In particular, we show that proper application of VC-bounds using correct estimates of VC-dimension provides accurate modeling of double descent curves, for various classifiers trained using stochastic gradient descent (SGD), least squares loss and standard SVM loss. The proposed VC-theoretical explanation provides many additional insights on generalization performance during first descent vs. second descent, and on the effect of statistical properties of the data on the shape of double descent curves.

Next, we briefly review VC-theoretical concepts and results necessary for understanding generalization performance of all learning methods based on minimization of training error [8, 9, 10, 11]:

1. Finite VC dimension provides *necessary* and *sufficient* conditions for good generalization.
2. VC theory provides analytic bounds on (unknown) test error, as a function of training error, VC dimension and the number of training samples.

Clearly, these VC-theoretic results contradict an existing consensus view that VC-theory cannot account for generalization performance of large DL networks. This disagreement results from misinterpretation of basic VC-theoretical concepts in DL research. These are a few examples of such misunderstanding:

– A common view that VC-dimension grows with the number of parameters (weights), and therefore, 'traditional measures of model complexity struggle to explain the generalization ability of large artificial neural networks' [3]. In fact, it is well known that VC-dimension can be equal, or larger, or smaller, than the number of parameters [8, 11].

– Another common view is that 'VC dimension depends only on the model family and data distribution, and not on the training procedure used to find models' [12]. In fact, VC dimension does not depend on data distribution [8, 11, 13]. Furthermore, VC dimension certainly depends on SGD algorithm [8, 11].

For classification problems, VC theory provides analytic generalization bounds for (unknown) Prediction Risk (or test error), as a function of Empirical Risk (or training error) and VC-dimension ($h$) of a set of admissible models, aka approximating functions. That is, for a given training data set (of size $n$), VC generalization bound has the following form [8, 9, 10, 11]:

$$R_{tst} \leq R_{trn} + \frac{\varepsilon}{2} \left( 1 + \sqrt{1 + \frac{4R_{trn}}{\varepsilon}} \right) \tag{1}$$

$$\text{where } \varepsilon = \frac{a_1}{n} \left( h \left( \ln \left( \frac{a_2 n}{h} \right) + 1 \right) - \ln \frac{\eta}{4} \right), \eta = \min \left( \frac{4}{\sqrt{n}}, 1 \right)$$

This VC bound (1) holds with probability $1 - \eta$ (aka *confidence level*) for all possible models (functions) including the one minimizing the training error ($R_{trn}$). The second additive term in (1), called the *confidence interval*, aka *excess risk*, depends on both the empirical risk (training error) and VC dimension ($h$). This bound describes the relationship between training error, test error and VC-dimension, and it is often used for *conceptual understanding* of model complexity control, i.e. understanding the effect of VC-dimension on test error. Application of this bound for accurate modeling of double descent curves requires:

– *Selecting proper values of positive constants $a_1$ and $a_2$.* The worst-case values $a_1 = 4$ and $a_2 = 2$, provided in VC-theory [8, 9] result in VC-bounds that are too loose for real-life data sets [11]. For classification problems, we suggest using the values $a_1 = a_2 = 1$, used for all empirical results presented in this paper.

– *Analytic estimate of VC-dimension.* For many learning methods, including DL, analytic estimate of VC-dimension are not known. For example, for SGD style algorithms, the effect of various heuristics (e.g., initialization of weights, etc.) on VC-dimension is difficult (or impossible?) to quantify analytically.

Note that VC bound (1) provides conceptual explanation of both first and second descent. That is, first descent corresponds to minimizing this bound when training error is non-zero [8, 9, 11]. Whereas

second descent corresponds to minimizing this bound when training error is kept at zero, using a set of models having small VC-dimension. This can be shown by setting the training error in bound (1) to zero, resulting in the following simplified bound for test error during the second descent:

$$R_{tst} \leq \varepsilon, \qquad \text{where } \varepsilon = \frac{h}{n} \left( \ln \left( \frac{n}{h} \right) + 1 \right) \tag{2}$$

More formally, since VC bound (1) depends only on two factors, training error and VC-dimension, there are two different strategies for minimizing this bound [11]:

- For a set of functions (models) with fixed VC-dimension, reduce the training error. This strategy leads to well-known classical bias-variance trade-off aka 'first descent';
- For small (fixed) training error, minimize the VC-dimension. This strategy corresponds to second descent, when training error is zero.

These two strategies correspond to *different methods* for controlling VC-dimension, that have been known long before DL. For example, the second strategy corresponds to margin maximization in SVM. Traditional learning methods typically implement a *single strategy*, whereas practitioners in DL observed the effect of *both strategies* when varying a single hyper parameter, such as network size or the number of epochs.

The main technical reason for misapplication of VC-theory, besides misinterpretation of VC-dimension, is that VC bound (1) remains virtually unknown in the DL community. That is, all technical arguments suggesting that VC-theory is unable to explain double descent, are based on analysis of *uniform convergence bounds* [1, 3, 14, 15, 16, 17]. In such bounds, the confidence interval term, aka the excess error, is of the order $\mathcal{O}\left(\sqrt{h/n}\right)$, i.e. it does not depend on empirical risk (training error). However, VC-theory also provides more accurate *uniform relative convergence* bounds, such as VC-bound (1), presented in [8, 9, 10, 11], where the confidence interval term also depends on the training error. So, whereas it is true that uniform convergence bounds cannot explain double descent, it can be fully explained by uniform relative convergence bounds.

Training of DL networks is based on stochastic gradient descent (SGD), which incorporates several heuristic rules to ensure that the norm of weights remains small. These rules include: initialization of weights to small random values and re-normalization during training. Consequently, for large DL networks, the model complexity is determined by the norm of weights, rather than the number of weights (parameters). Further, this dependence of VC-dimension on the training algorithm helps explain why theoretical estimates of VC-dimension based only on network topology have found little practical use [11].

## 2 Application of VC Bounds for Modeling Double Descent

This section presents a VC-theoretical explanation of double descent for classification, for a single-layer network shown in Figure 1. The same network setting was used for analysis of double descent in recent papers [1, 18, 19, 20]. In this network, a classifier is estimated in two steps:

- First, input vector $\boldsymbol{x}$ is encoded using N nonlinear features. Commonly, *random features* (weak features) are used, such as random ReLU or Random Fourier Features (RFF);
- Second, a linear model is estimated in this N-dimensional feature space.

This simplified setting enables VC theoretical analysis of double descent, because the analytic estimates of VC-dimension are known. That is, since the network output is formed as a linear combination of N features, analytic estimate of VC-dimension for linear hyperplanes $f(\boldsymbol{z}, \boldsymbol{w}) = (\boldsymbol{w}\boldsymbol{z}) + b$ is known [8, 9, 10]:

$$h \leq \min \left( ||\boldsymbol{w}||^2, N \right) + 1 \tag{3}$$

This bound holds under the assumption that all training samples are enclosed within a sphere of radius 1, in **Z**-space. In summary, VC-dimension can be bounded by the input dimensionality (N), or by the norm of weights. These are two different mechanisms for controlling VC-dimension.

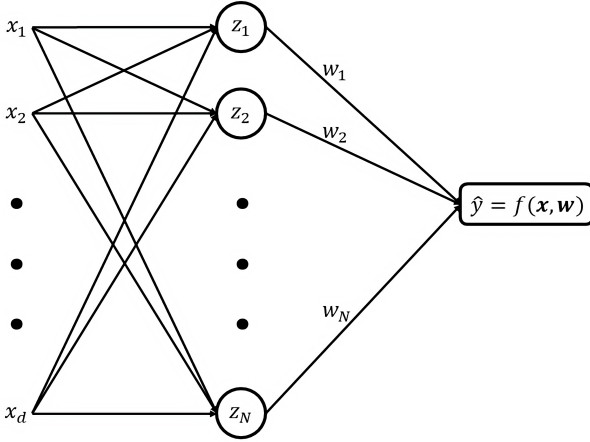

Figure 1: Single hidden layer network estimating a linear classifier in nonlinear feature space (**Z**-space).

128 The double descent phenomenon can be observed for various learning methods used to estimate
129 weights $w$ for the network structure in Figure 1. Next, we present empirical results showing the
130 application of VC bounds (1) and (3) for modeling double descent when network weights are
131 estimated using SVM or Least-Squares (LS) classifiers. For large networks trained using LS, when
132 N is larger than sample size ($n$), minimization of squared error is performed using pseudo-inverse,
133 which finds a solution corresponding to the minimization of the norm squared $||w||^2$.

134 In all experimental results in this section, double descent is observed when the network size (N) is
135 gradually increased. Specifically, according to analytic bound (3):

- When network size (N) is small, the VC-dimension initially grows linearly with N. This
  corresponds to the first descent, or traditional bias-variance trade-off.

- For overparameterized networks (large N), VC-dimension is controlled by the norm squared
  of weights, leading to second descent.

140 We use two types of random nonlinear features [1, 21], ReLU and RFF. Random ReLU features are
141 formed as:
$$\boldsymbol{Z}_i = \max\left(\langle \boldsymbol{v}_i, \boldsymbol{X} \rangle, 0\right), \quad i = 1, ..., N$$

142 where random vectors $\boldsymbol{v}_1, \ldots, \boldsymbol{v}_N$ are sampled uniformly from the range [-1,1]. Random Fourier
143 Features (RFF) are formed as:

$$\boldsymbol{Z}_i = \exp\left(\sqrt{-1}\langle \boldsymbol{v}_i, \boldsymbol{X} \rangle\right), \quad i = 1, ..., N$$

144 Where random $\boldsymbol{v}_1, \ldots, \boldsymbol{v}_N$ are sampled from Gaussian distribution with standard deviation $\sigma = 0.05$.
145 In all experiments, input ($\boldsymbol{x}$) values were pre-scaled to [0, 1] range, for both training and test data.

146 Following the nonlinear mapping $\mathbf{X} \rightarrow \mathbf{Z}$, all $\boldsymbol{z}$-values are re-scaled to [-1, 1] range. Such re-scaling
147 is performed to satisfy the condition for bound (3), stating that all training samples in **Z**-space should
148 be enclosed within a sphere of radius 1.

149 Training samples ($\boldsymbol{z}$, $y$) are used to estimate a decision boundary in **Z**-space. Two different methods
150 (LS and SVM classifiers) are used for estimating linear decision function $f(\boldsymbol{z}, \boldsymbol{w}) = (\boldsymbol{w}\boldsymbol{z}) + b$ from
151 training data, in order to show double descent curves for two *different loss functions*, LS and SVM
152 loss. For SVM modeling, the regularization parameter $C$ is set to 64 in all experiments. Empirical
153 test error is estimated using an independent test set.

154 Most empirical results in this paper were obtained for MNIST digits adapted for binary classification
155 (digit 5 vs 8), where digits are grey-scale images of size 28x28. The training set size $n = 800$, and test
156 set size is 2,000. We have also used other data sets for modeling and observed similar results. See
157 Appendix for additional results.

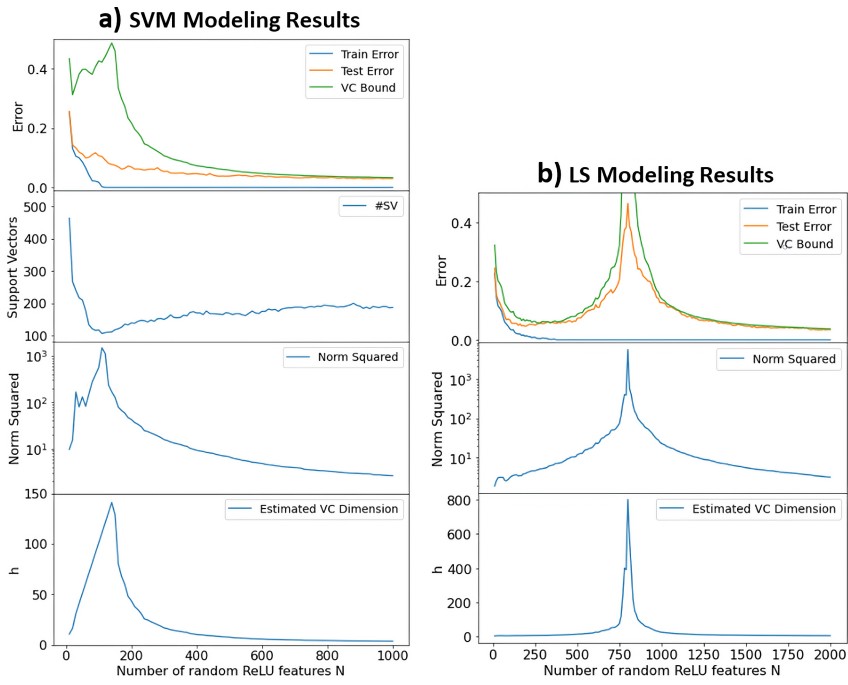

Figure 2: Application of VC-bounds for MNIST digit 5 vs 8 data set using random ReLU features.

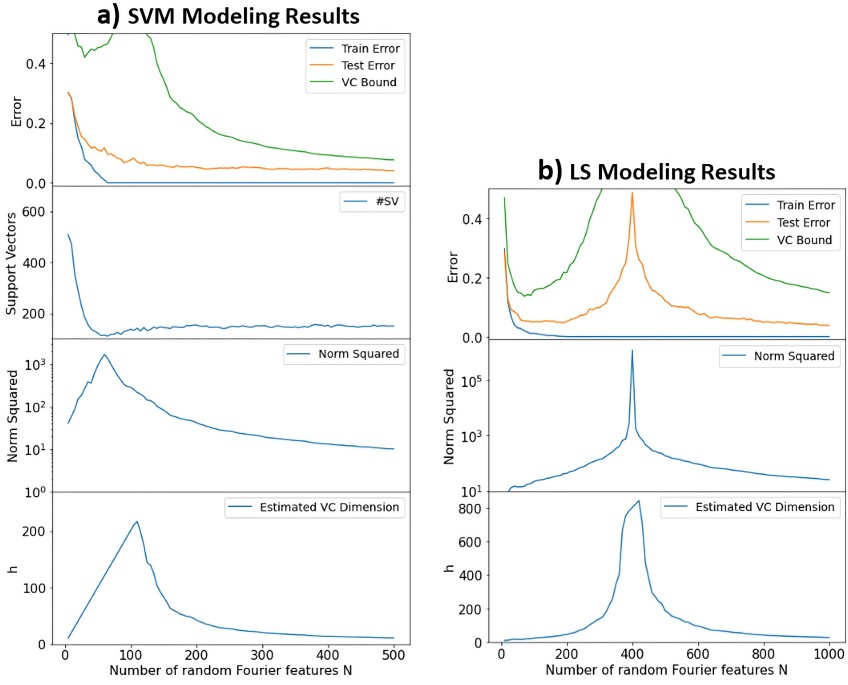

Figure 3: Application of VC-bounds for MNIST digit 5 vs 8 data set using RFF.

Empirical results in Figures 2 and 3 show application of VC bounds to modeling MNIST data. They show:

– Empirical training and test error curves, as a function of N ( the number of nonlinear features), along with VC-theoretical estimate of test error obtained via bounds (1) and (3). These curves show that analytic VC-bounds can explain (and predict) double descent;

163          – The 'norm squared' of estimated linear model, as a function of the number of features N;

164          – For SVM, we also show the number of support vectors for trained SVM model.

165 Modeling results for random ReLU and RFF features are similar, so we only comment on results in
166 Figure 2:

167          – *For small N*, VC-dimension grows linearly with N for SVM method. Empirical results show
168            that first descent error curves can be explained by VC-bound (1), because the minimum of
169            VC-bound closely corresponds to the minimum of test error. This can be clearly seen for LS
170            classifier, and less obvious for SVM.

171          – *For large N*, VC-dimension is controlled by the norm squared, according to bound (3). These
172            results show that second descent can be explained by VC-bound (1), for both SVM and LS
173            learning methods.

174 Whereas empirical results for both SVM and LS in Figure 2 are qualitatively similar, their double
175 descent curves show different values of *interpolation threshold* N* (where the training error reaches
176 zero). For SVM, the value N* $\approx$ 100 is achieved when the number of features equals the number of
177 support vectors. For LS classifier, the interpolation point N* $\approx$ 800 is achieved when the number of
178 features equals the number of training samples.

179 The dependence of test error on the norm of weights in large networks has been known to practitioners,
180 and some limited theoretical explanation is provided in [1, 22, 23]. For example, [1, 22] suggest
181 that minimum norm provides inductive bias by favoring models with higher degree of smoothness.
182 However, these papers do not mention VC bounds that clearly relate the VC-dimension to the norm
183 of weights, and explain generalization performance for linear classifiers.

184 The dependence of interpolation threshold N* on training sample size for LS classifiers has also
185 been observed in the DL literature. However, in the absence of sound theoretical framework for
186 double descent, interpretation of this empirical dependency leads to convoluted explanations. For
187 example, Nakkiran, et al. [12] investigated the effect of varying the number of training samples on
188 test error, for a fixed-size DL network. In particular, they observed two double descent error curves
189 for the same network trained using smaller and larger size training data, showing that during second
190 descent, near interpolation threshold, the test error for a network trained with larger data set is worse
191 than for the same network trained on smaller data set. This phenomenon was called 'sample-wise
192 non-monotonicity', and a new theory was proposed for explaining regimes where 'increasing the
193 number of training samples actually hurts test performance'. However, this phenomenon has a simple
194 VC-theoretical explanation, as explained next. Note that for LS classifiers during second descent the
195 shape of the 'norm squared' closely follows the shape of test error, according to VC bound (2), as
196 evident in Figures 2 and 3. Since for LS classifiers the interpolation threshold is given by training
197 sample size, there is a region near interpolation threshold, where VC-dimension for a smaller training
198 size is smaller than for a larger training size. According to VC-bound (2), in this region of second
199 descent we can expect a smaller (better) test error for a smaller training size.

200 VC theoretical framework can also help to understand the effect of statistical characteristics of training
201 data on generalization curves. Next, we present empirical results demonstrating the effect on noisy
202 data on the shape of double descent curves, along with their VC theoretical explanation. For these
203 experiments, we use a single-layer network trained using SVM and LS classifier using random ReLU
204 features. We use digits data with corrupted class labels. The training set size is 800 (400 per class),
205 and the test set size is 2,000. Figure 4 shows the effect of noise level on the shape of double descent
206 curves, for SVM and LS classifiers. Results for both SVM and LS models show double descent
207 curves, but their shape is different. For the SVM model estimated using 'clean' data (0% label noise),
208 there is no visible first descent at all, but for noisy data we observe both first and second descent.
209 For the LS model, we clearly observe first and second descent for both clean and noisy training data.
210 For LS curves, the interpolation threshold ($\approx$ training size 800) is the same for different noise levels,
211 but for SVM the value of interpolation threshold increases with noise level in the data. This can
212 be explained by noting that for SVM, the interpolation threshold is reached when the training data
213 becomes linearly separable (in nonlinear feature space, or **Z**-space in Figure 1). Therefore, for SVM
214 the interpolation threshold is given by the number of support vectors needed to separate training
215 data. For noisy data, a SVM model requires a larger number of support vectors, resulting in larger
216 interpolation threshold.

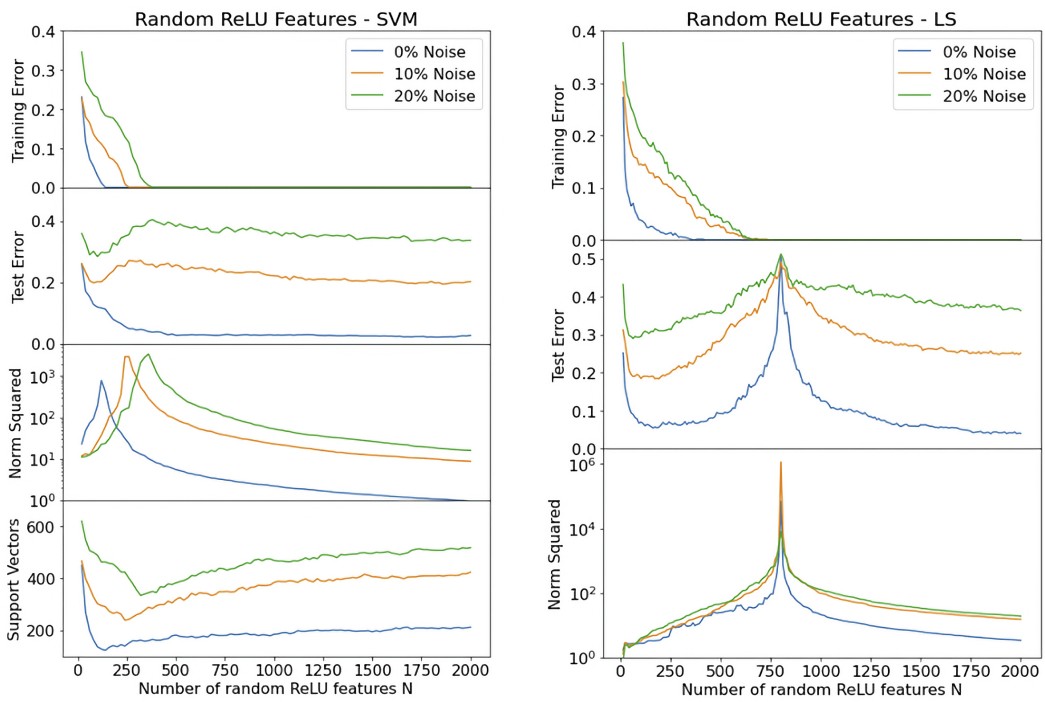

Figure 4: Effect of noise on double descent for MNIST data, for SVM and LS classifiers using random ReLU features.

The ability to generalize for noisy data can be explained by noting that during second descent VC bound (2) depends only on VC-dimension (the norm of weights). With increasing noise (in the data), the norm of weights increases, resulting in degradation of test error and flattening of second descent test error curve (as evident in Figure 4, for both SVM and LS).

Empirical results in Figure 4 also show that for noisy data, generalization performance during second descent degrades, relative to optimal first descent model. This is contrary to the popular view that DL networks usually provide superior generalization performance during second descent [1, 3, 4, 12].

We suspect that superior performance during second descent, reported in the DL community, can be explained by using large and 'clean' data sets (common in Big Data). For such training data sets (of size $n$), generalization performance during second descent is likely to be good, because the VC bound (2) on test error depends only on the ratio of VC-dimension to sample size $(h/n)$.

## 3 Modeling Double Descent for Fully Connected Multilayer Networks

Empirical results for a simplified network setting in Section 2 provide insights for generalization performance of over-parameterized multilayer networks. Such general DL networks use SGD training that keeps the norm of weights small, so that the model complexity is determined by the norm of weights, rather than the number of weights (parameters). However, direct application of analytic VC bounds to modeling double descent may be tricky, because we need to address two challenging research issues:

1. How to estimate VC-dimension for general DL networks, where analytic estimates do not exist;

2. Understanding design choices for setting multiple 'tuning' parameters, such as network width, number of training epochs, weight initialization, etc. All of these hyperparameters can be used to control the VC dimension of DL networks. Double descent curves show dependence of training and test error on a *single complexity parameter*, when all other tuning parameters are preset to 'good' values.

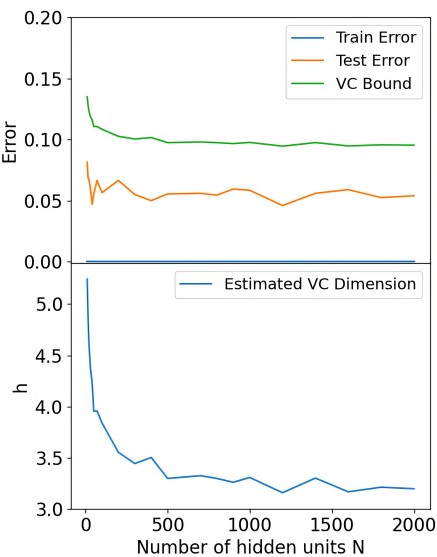

Figure 5: Modeling results for second descent as a function of network width (N).

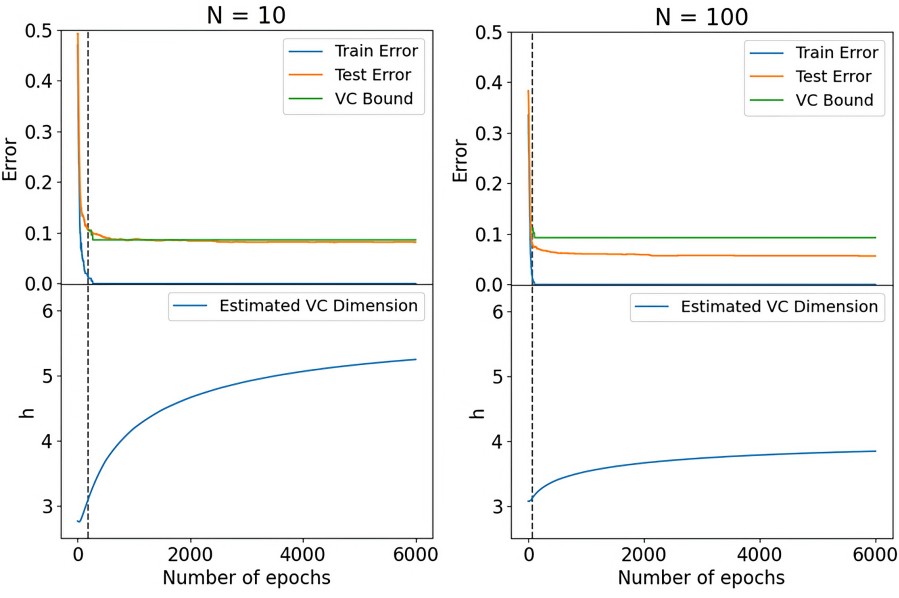

Figure 6: Second descent as a function of the number of epochs, for N = 10 and N = 100.

For these reasons, the application of analytic VC bounds to general DL networks is difficult (or impossible?). However, it can still be done for restricted and well-defined network settings. In this section, we consider a fully connected network with a single hidden layer, as in Figure 1, where the network weights in *both layers* are estimated during training via SGD. In this case, $z$-features are *adaptively estimated* from training data, in contrast to *fixed* random features used earlier in Section 2.

Let us consider two factors (hyperparameters) controlling complexity of such networks trained via SGD: the number of hidden units (N), and the number of training epochs. Empirical results showing double descent curves as a function of these two factors have been extensively reported in DL literature [1, 12]. However, our purpose here is not to replicate such double descent curves, but to explain them using VC bounds (1) and (3). In order to make it possible, we have to specify network setting where VC-dimension can be approximately estimated. Therefore, we only consider modeling for second descent, where training error is kept very small (or zero), and test error is bounded by (2). This can be achieved when the number of hidden units N is large, or the number of training epochs is

large. So, our experiments intend to show application of VC bounds *only in such restricted settings*, where varying one complexity factor (for example, N) has an effect on VC-dimension and test error, according to bound (2).

Further, in order to estimate VC dimension, we hypothesize that under *such restricted settings*, during second descent, the norm of weights in the output layer can be used to approximate the VC-dimension of a neural network. The reasoning (behind this hypothesis) is that for such restricted settings the training error is zero, so a linear decision boundary in **Z**-space should have a large separation margin, i.e. VC dimension is controlled mainly by the norm of weights in the output layer. This assertion appears to be supported by experiments for fully connected networks, trained on several different data sets.

Next, we present empirical results for MNIST digits data, under the following experimental setting:

- 200 training and 2,000 test examples (of digits 5 and 8);
- Fully connected network using random ReLU activation function in hidden units [24];
- Training using SGD with learning rate 0.001 and momentum 0.95. The learning rate is reduced by 10% for every 500 epochs. Batch normalization is used during training.
- Weights initialized, prior to training, using Xavier uniform distribution, following [25].

Our design choices for SGD implementation mainly follows earlier studies [1, 25].

Figure 5 shows modeling results for second descent mode, as a function of the number of hidden units N (the number of epochs is set to 6000 in all experiments). The top part shows empirical training and test error curves, and the VC-bound on test error, that closely approximates empirical test error. The bottom part of the figure shows the VC-dimension, estimated as the norm squared of the output layer weights (for trained network). Figure 6 shows modeling results for second descent mode, as a function of the number of epochs (for networks with N = 10 and 100 hidden units). Note that VC-bound and VC-dimension can be reliably estimated only in second descent mode, when training error is close to zero. This region, where training error is smaller than 1%, is indicated by dotted vertical line. These results show that in the region where training error is very small, increasing the number of epochs results is a small increase in VC-dimension and a slight decrease of training error. This is a particular form of memorization-complexity trade-off, implicit in VC bound (1), when training error is very small (close to zero).

These results demonstrate applicability of VC bounds for modeling second descent in fully connected multilayer networks. In addition, we can see the effect of each complexity parameter (network size N and the number of epochs) on VC-dimension. This can be used for ranking tuning parameters, according to their ability to control VC-dimension of DL networks during second descent.

## 4   Summary

This paper provides a VC-theoretical explanation of 'double descent' in multilayer networks. We show that for simplified network structures where VC-dimension can be analytically estimated, VC generalization bounds can be applied directly to the model and predict a double descent phenomenon.

VC-theoretical framework is helpful for improved understanding of empirical results observed in DL, such as: the effect of various heuristics on generalization, relative performance of first and second descent, etc. Another important VC-theoretical insight is that during second descent VC-dimension depends on the norm of weights. According to VC-theoretical explanation, second descent occurs when zero training error is achieved using an estimator having small VC-dimension, i.e. small norm of weights. This phenomenon is general, and it does not depend on a particular training algorithm or on a chosen parameterization (such as multilayer network). Therefore, double descent can be observed for other learning methods, such as SVM estimators, and not only for DL networks trained by SGD algorithm.

Empirical results presented in this paper contradict the consensus opinion that VC-theory cannot explain prediction performance of neural networks. Possible future work in this area may investigate VC theoretical modeling of double descent for *low-dimensional data*, and also for regression problems. Note that for regression problems VC-theoretical bounds have a different form [8, 9, 10, 11], and these bounds have not been previously used for modeling second descent.

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
