# A   Appendix

This Appendix includes additional empirical results for modeling double descent in a single-layer network shown in Figure 1 in the main paper. First set of results investigates the choice of theoretical constants $a_1$ and $a_2$ in VC bound (1) reproduced below:

$$R_{tst} \leq R_{trn} + \frac{\varepsilon}{2}\left(1 + \sqrt{1 + \frac{4R_{trn}}{\varepsilon}}\right)$$

$$\text{where } \varepsilon = \frac{a_1}{n}\left(h\left(\ln\left(\frac{a_2 n}{h}\right) + 1\right) - \ln\frac{\eta}{4}\right), \eta = \min\left(\frac{4}{\sqrt{n}}, 1\right)$$

VC theory [1, 2] specifies their range and provides the values corresponding to pessimistic assumptions (about unknown data distributions):

  – the range [0, 4] for $a_1$ and [0,2] for $a_2$.

  – worst-case correspond to values $a_1 = 4$ and $a_2 = 2$.

These worst-case values result in upper bounds that are too crude for real-life data sets. Therefore, for low-noise data sets in the main paper, we used values $a_1 = 1$ and $a_2 = 1$. However, for noisy data we should use larger values. The choice of proper values for these theoretical constants for noisy data is discussed next, using LS classifiers with ReLU features, for digits data (the same 5 vs. 8 data set as in the main paper). For this data set, we introduce noise by using corrupted class labels. That is, we consider 3 data sets, with 0% noise, 5% noise and 10% noise, where 0% noise refers to original 'clean' data set (used in the main paper). Figure A.1 shows modeling results for data set with 5% noise, using VC bounds with $a_1 = 1$ and $a_2 = 1$. These results show that VC-bounds underestimate empirical curves for both first and especially second descent. However, using values $a_1 = 3$ and $a_2 = 1$ results in practical VC bounds that provide accurate modeling of double descent for this data, at various noise levels. See empirical results in Figure A.2.

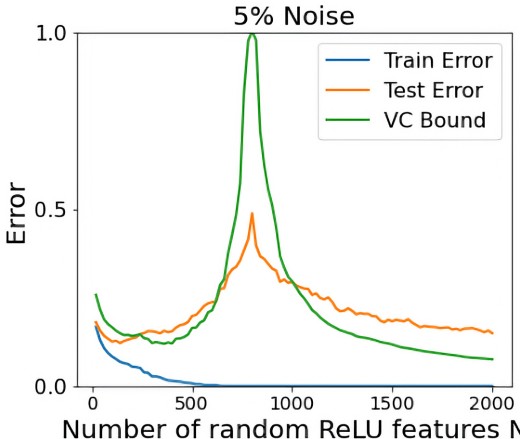

Figure A.1: Modeling double descent for digits data with 5% label noise, using values $a_1 = 1$ and $a_2 = 1$.

Next, we show experimental results for the same digits data set, when images are corrupted by random Gaussian noise. Here, the noise level is given by standard deviation of the Gaussian noise $\hat{\sigma} = 0, 0.1, 0.2$. Empirical results in Figure A.3 show that VC bounds (with values $a_1 = 3$ and $a_2 = 1$) provide accurate modeling of double descent, at various noise levels.

We can conclude that these values $a_1 = 3$ and $a_2 = 1$ provide robust VC theoretical modeling of double descent for noisy data. For example, Figure A.4, shows modeling double descent for CIFAR10 data set (cat vs automobile), extracted from CIFAR10 data base. This data set has 800 training samples and 2,000 test samples. Modeling results are obtained for the network with random ReLU features, trained using LS classifier.

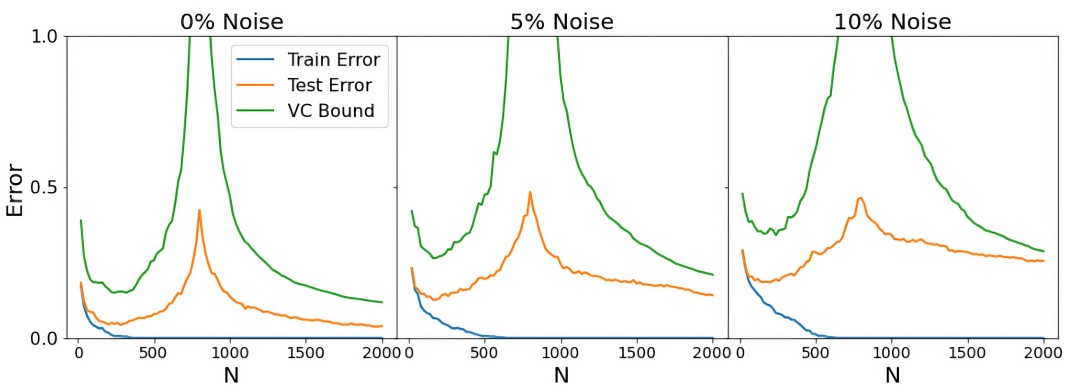

Figure A.2: Modeling double descent for digits data with corrupted class labels, using values $a_1 = 3$ and $a_2 = 1$.

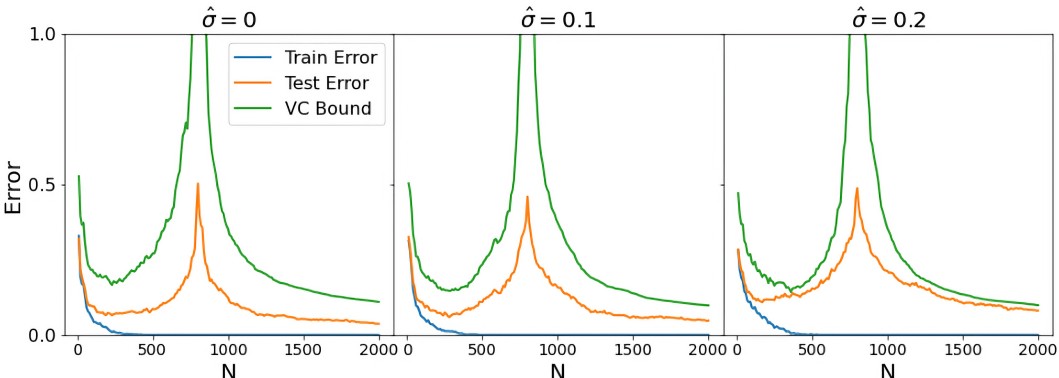

Figure A.3: Modeling double descent for digits data with corrupted pixel, using values $a_1 = 3$ and $a_2 = 1$.

30 Last set of results shows the effect of varying the number of training samples on test error, for a
31 fixed-size network. This setting was used in [3]. Results in Figure A.5 show modeling double descent
32 for digits data set, using fixed-size network with N=500 and N=1500. These results are obtained for
33 the network with random ReLU features and are trained using LS classifier with $a_1 = 1$ and $a_2 = 1$.
34 They show very accurate modeling of double descent using VC-bounds, under this setting.

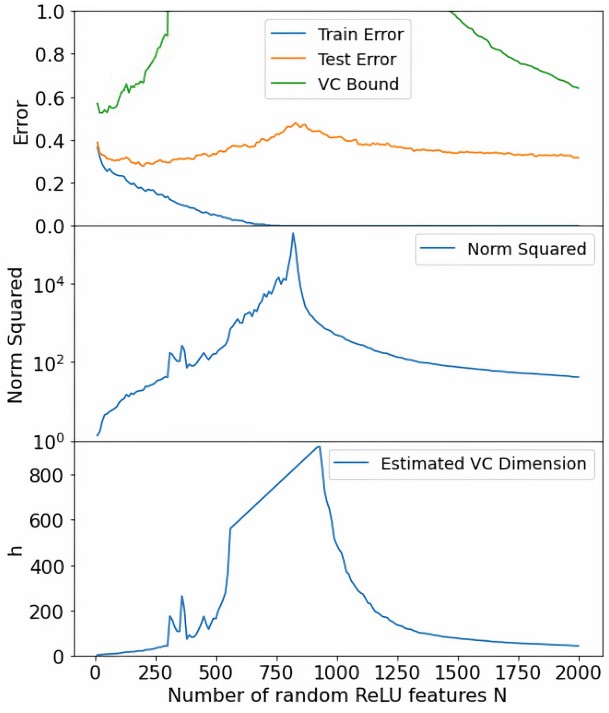

Figure A.4: Modeling double descent for CIFAR data (Cat vs Automobile), using values $a_1 = 3$ and $a_2 = 1$.

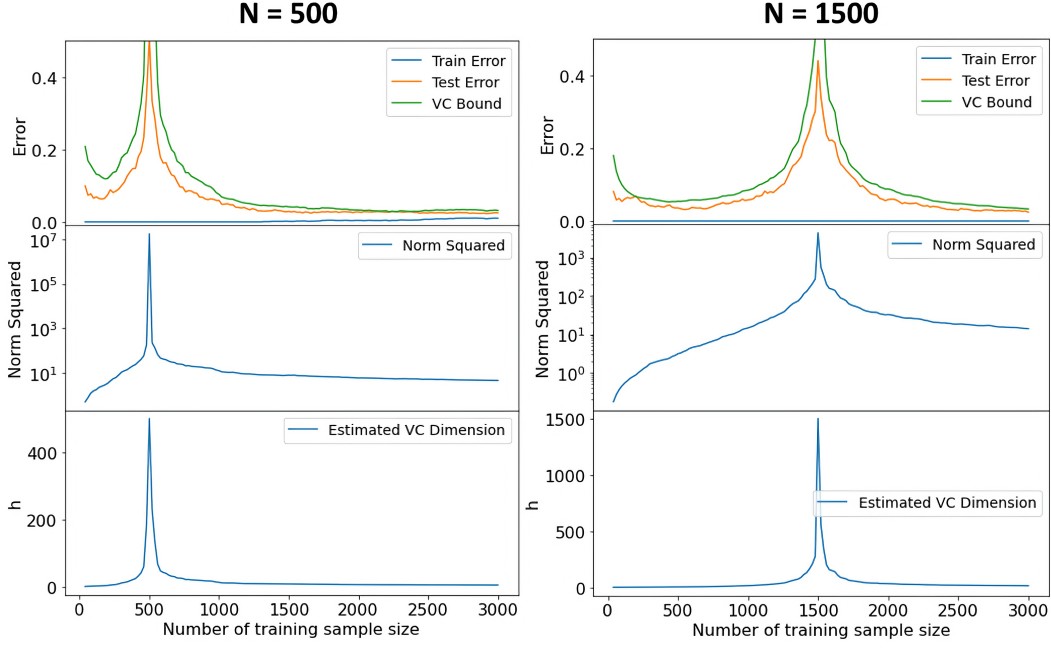

Figure A.5: Modeling the effect of varying the number of training samples on test error, for a fixed-size network.