# OpenReview forum: "VC Theoretical Explanation of Double Descent"
_NeurIPS.cc/2022/Conference — NeurIPS 2022 Submitted_

### Official Review · Reviewer_Zr4x · 2022-06-20

**Rating:** 7
**Confidence:** 5
**Soundness:** 3 good
**Presentation:** 3 good
**Contribution:** 3 good

**Summary:**

This paper presents VC-theoretical analysis of double descent and shows that it can be fully explained by classical VC generalisation bounds. Theoretical and experimental results support the analysis.

**Questions:**

It is now always true that the norm of the weight goes down with the number of parameters (see e.g., Belkin et. al. and its experiment).
How you can deal with this cases?
Can't you show similar results using Rademacher or Local Rademacher Complexity?
[Belkin] https://arxiv.org/pdf/2105.14368.pdf

**Limitations:**

It is now always true that the norm of the weight goes down with the number of parameters (see e.g., Belkin et. al. and its experiment).
[Belkin] https://arxiv.org/pdf/2105.14368.pdf

**Strengths And Weaknesses:**

Strengths
- approach is sound and theoretical grounded
- explanation of this phenomena by means of well known theory

Weaknesses
- there is actually nothing new theoretically
- experimental result is limited

---

> ### Author Response · Authors · 2022-08-01
> **Response to Reviewer Zr4x**
>
> We thank the reviewer for reviewing our paper and their overall positive assessment. Please see below our responses to the questions.
>
> - During second descent, the norm of weights always goes down, as the number of features (N) is increasing, because for larger N we approach optimal kernel solution, as explained in [Belkin et al, 2018; Belkin, 2021].
> - Rademacher complexity is used for data-dependent bounds – they cannot be used in our paper, assuming unknown distributions. This direction can be explored in the future work.
>
>
> Reference
>
> Belkin, M., Ma, S. & S, Mandal. (2018) To understand deep learning we need to understand kernel learning, In Proc. ICML.
>
> Belkin, M. (2021) Fit without fear: remarkable mathematical phenomena of deep learning through the prism of interpolation, Acta Numerica.

---

### Official Review · Reviewer_vA1k · 2022-06-27

**Rating:** 3
**Confidence:** 3
**Soundness:** 2 fair
**Presentation:** 1 poor
**Contribution:** 3 good

**Summary:**

The authors present an alternative explanation for the “double-descent” phenomenon [1] sometimes observed in training classifiers as their capacity is increased. They claim that a VC-theoretic analysis can explain this behaviour, in contrast to the claims of prior work.

Their analysis is based on two changes to prior analysis: the use of relative-deviation VC bounds (which interpolate between slow and fast rates as the empirical risk tends to zero) rather than the slow-rate uniform deviation bounds commonly used; and the use of norm-based bounds on the VC dimension for larger models, for example the bound $V_C \le \min(\operatorname{dim}_\text{input}, \|w\|_2^2) + 1$ for linear classifiers.

The behaviour of this VC bound is compared directly to the training and testing error in settings seen in double-descent literature, such as linear classification based on fixed random Fourier or ReLU features, and one-hidden-layer ReLU networks using a hypothesised estimate for VC dimension in these settings.

[1] Reconciling modern machine learning practice and the bias-variance trade-off. Belkin, Hsu, Ma & Mandal.

**Questions:**

1. In l.76 you mention that you use alternative constants in the VC bound to that given by the theory. Are these just chosen in an ad-hoc way or justified by any theoretical development?

2. In l.146 you say that you rescale z values to $[-1, 1]$ to meet the condition for equation (3). But should this not be rescaling to $|z|_2 < 1$ instead?

**Limitations:**

I do not feel the authors discuss enough the limitations of the assumptions they use in obtaining their empirical results.

Societal impact: N/A this is a theoretical work.

**Strengths And Weaknesses:**

### Strengths

**Originality**: I am not too familiar with the literature since the early "double descent" papers, but it seems to me that the better VC bounds applied in this paper are new in this setting, although not new in general.

**Central idea**: I think the central (implied) idea of the paper is that the double descent phenomenon is observed because once we reach the interpolation threshold, we are minimising a norm instead of the error. This is actually mentioned already in [1], but the fact that this can be explained (at least in some cases) already through VC theory is a significant idea.

### Weaknesses

**Clarity**: This is one of the biggest shortcomings of this work. There are many grammatical mistakes, and I found myself flicking back and forward to look for explanations or definitions. The subject material is not particularly technical so I feel the authors could do much better here.

**Rigour**: This is another major shortcoming. A number of ad-hoc assumptions seem to be made, for example the way the authors assume that the VC dimension of a shallow neural network can simply be approximated by the final layer norm, or how it appears (see question) that the authors change the constants in the VC bound, because the original constants are "too loose for real-life data sets".

---

> ### Author Response · Authors · 2022-08-01
> **Response to Reviewer vA1k**
>
> Thank you for reviewing our paper. Please see below our responses.
>
> \\( \\normalsize{\\text{Regarding to Clarity}} \\)
>
> We agree and will perform thorough proofreading of the revised paper.
>
>
> \\( \\normalsize{\\text{Regarding to Rigour}} \\)
>
> We partially agree with this criticism, regarding approximating VC-dimension of a wide network (by the norm of weights of the output layer), noting the original paper provides specific assumptions and restricted settings (i.e., second descent mode, large N) under which this approximation can be used. However, we can certainly provide additional clarifications. Our assumption is that during second descent mode, VC-dimension of a wide network can be approximated by the norm of weights (of the output layer). This is based on the growing understanding that training sufficiently wide single-layer networks (using SGD with squared loss) is similar to training a linear classifier (via pseudo-inverse calculation). See [Belkin 2021] for mathematical explanation and discussion of this phenomenon, where it is called ‘transition to linearity’. Also, [Bietti and Bach, 2021] argue that very wide fully-connected multilayer networks have essentially the same approximation properties as their “shallow” two-layer counterparts, effectively implementing standard kernel machines.
>
> Under this assumption, we can interpret learning in wide DL networks during second descent in the following fashion:
> - learning amounts to finding a solution minimizing the norm of weights in the output layer (linear estimation problem)
> - the output layer weights represent many weak nonlinear features formed by previous layers. (particular mechanism used to form such nonlinear features is not critical for generalization, i.e. one can use different types of activation functions in neural networks, or different kernels in SVM).
> According to such interpretation, minimization of the norm of weights in the output layer is mainly responsible for generalization for both SVM loss and Least Squares (LS) loss, and this justifies our approximation for VC dimension. Further, according this view, we can expect that for wide networks (large N), generalization performance during second descent will be similar for both linear networks and nonlinear networks (with same large N). In fact, this view is confirmed by empirical results for estimated VC-dimension for linear network and nonlinear network (both using the same ReLU features), for large N values. For example, for network size N= 2,000, estimated norm_squared of the weights in the output layer is 3.3 for linear network and 2.3 for nonlinear network. This supports our assumption for using the norm of weights in the output layer, as a proxy for VC-dimension, for wide networks during second descent.
>
> The revised paper will include additional justification for this approximation, along with additional references:
>
> Belkin, M. (2021) Fit without fear: remarkable mathematical phenomena of deep learning through the prism of interpolation, Acta Numerica.
>
> Bietti, A. & Bach, F. (2021) Deep Equals Shallow for ReLU Networks in Kernel Regimes, in Proc. ICLR.
>
> \\( \\normalsize{\\text{Regarding to Question 1}} \\)
>
> Classical VC-theory provides the form of VC bounds, up to the value of (positive) theoretical constants, in the range 0<a1<4 and 0<a2<2. The worst-case values a1=4 and a2=2, corresponding to worst-case “heavy-tailed” distributions, are usually assumed in VC-literature. For practical applications, these values may need to be tuned, depending on the knowledge about application data. However, we certainly do not suggest that the values should be tuned to particular data set. In the revised paper, we will present all modeling results (in the main paper) using a1=3, a2=1. Additional discussion regarding more refined selection of a1=a2=1 values will also be presented in Appendix. The general rule is that for low-noise data sets, such as MNIST, using values a1=3 and a2=1 results in more accurate bounds (when this a priori knowledge is available)
>
> \\( \\normalsize{\\text{Regarding to Question 2}} \\)
>
> This is a valid point, we performed coordinate-wise rescaling, as a short cut, instead of finding the minimum sphere enclosing all z-values. Finding the minimum sphere requires solving a separate quadratic optimization problem, for each training data set – this can be done in the future. However, even this simplified rescaling, as implemented in the paper, is technically correct, because it satisfies VC-theoretical assumptions for analytic estimate (3). For separable class distributions, such as digits data used in our paper, the simplified rescaling provides solution that is likely to be similar to the minimum enclosing sphere. Another reason for using simple rescaling (in our paper) is that it is commonly used for re-normalization of multivariate inputs (such as individual pixels of an image) commonly used for training in DL.

---

### Official Review · Reviewer_8Fpz · 2022-07-26

**Rating:** 2
**Confidence:** 5
**Soundness:** 1 poor
**Presentation:** 2 fair
**Contribution:** 1 poor

**Summary:**

The paper empirically investigates whether relative VC bounds can
explain the double descent phenomenon. To this end, random features
of size $N$ are created and then a (regularized) least squares
approach in the feature space is performed. For the resulting,
empirically observed weights, the relative VC bounds (among other
quantities) are plotted as function of $N$. The obtained curves reflect
the double descent phenomenon, at least for the unregularied case.
In addition, similar experiments are reported for NNs with one
hidden layer.

**Questions:**

Can you address the serious issues I identified? Or did I completely misunderstood your paper?

**Limitations:**

Unfortunately, the authors do not seem to be aware of the limitations of their approach outlined above.

**Strengths And Weaknesses:**

Unfortunately, the paper relies on a fundamental misunderstanding
of VC bounds. Indeed, in these bounds, one first needs to FIX a
class of hypotheses that is INDEPENDENT of the data samples, which
are then observed in a second step. For this reason, it does not make
sense to use an estimate on the VC dimension that is obtained AFTER
training (as done in the experiments via the norm of the weights), and for the same reason, the
discussion on the top of page 3 is entirely pointless.
In other words, unless you are able to prove in advance that the algorithm you consider produces a weight vector whose norm is not exceeding a certain a-priorily fixed threshold, you cannot apply these bounds. And of course, you need to plug-in this threshold in the bounds and not the observed norm.

The entire setup somewhat reminds me to the so-called "luckiness"
approach taken in the early 2000s, but there the goal was to first
derive solid new bounds, which is completely missing in this paper.

Finally, the authors idea to use relative bounds instead of the "usual" uniform bounds is also missing the point as both depend on $h/n$, where $h$ is (an upper bound of) the VC dimension and $n$ is the number of samples. The only major difference between these two type of bounds is that the uniform bounds depend on $h/n$ in a square-root fashion, whereas, at least in the ideal case, the relative bounds depend linearly on this fraction, see (2).

---

> ### Author Response · Authors · 2022-08-01
> **Response to Reviewer 8Fpz**
>
> We thank the reviewer for reviewing our paper. The reviewer raises two separate points, regarding applicability of VC bounds and the particular way of estimating VC dimension. Apparently, these points were raised for second descent regime, and they are addressed separately:
>
> - VC-bounds on test error, Eq (1) in our paper, hold for all models (functions) including the ones minimizing training error. In the paper, these bounds are used during second descent mode (when training error is reduced to zero). That is, we apply a given (over-parameterized) learning method to training data, and observe empirical test error and empirical (zero) training error.
>
> - the assertion that VC-dimension should not be estimated using training data is not correct. In fact, for delta margin hyperplanes, VC-dim depends on margin size, according to Eq (3) in the paper, and maximum margin size depends on particular training data set. Our use of bound (1) in conjunction with estimated VC-dimension (3) is common in classical VC theory – See, for example, Theorem 5.1 and its Corollary in [Vapnik 1999].
>
> We suspect that reviewer’s comments are caused by misunderstanding of the second strategy for minimizing VC-bound implemented as margin maximization (in SVM) or minimization of the norm of weights (via SGD training), during second descent mode. This strategy is described in VC-theory by defining a structure (ordered by the norm of weights) for each equivalence class (where the number of training errors is constant, e.g., zero training error during second descent). In this case, the structure is specified a priori (independent of data), and then its optimal element (minimum norm_squared of weights) is found using training data. The distinction between the two strategies for SRM (Structural Risk Minimization) is explained informally on top of page 3 in the original paper, along with reference [Cherkassky & Mulier 2007]. The reviewer can refer to [Cherkassky & Mulier 2007] or Vapnik’s books for more detailed explanations.
>
> We believe that reviewer’s critical comments reflect common misunderstanding of this second strategy in Deep Learning community, where it is regarded as a particular case of penalization, similar to ridge penalty. In fact, the usual penalization (using ridge penalty) implements the first SRM strategy (that usually works during first descent), but it is different from what is performed during second descent.
>
> Regarding the comment on uniform bounds, the reviewer correctly noted that the confidence interval (excess error) term is of the order O(h/n) for the relative bounds but of the order of SQRT (h/n) for the usual bounds used in machine learning literature. For small values, common in practical applications, this difference is huge. For example, for h/n = 0.1 (10% excess error) the usual bounds give SQRT (0.1) = 0.33 (33% excess error). We agree that this point was not clearly explained in the original paper and will be entered in the revised version.
>
> Note that our paper does not propose relative bounds – as the importance and advantages of using these bounds is clearly explained in all Vapnik’s books. However, previously these bounds have been used/ known only for conceptual explanation of generalization. Our main contribution is using these bounds for quantitative modeling of double descent using empirical data.
>
> Reference
>
> Vapnik, V. (1999) The Nature of Statistical Learning Theory, Springer.
>
> Cherkassky, V. & Mulier, F. (2007) Learning from Data, Second Edition, Wiley-Interscience.

---

### Meta-Review · Area_Chair_MKgX · 2022-08-27

**Recommendation:** Reject
**Confidence:** Certain

**Metareview:**

Having read the paper on my own, I am, like one of the reviewers, not convinced by the authors' approach: The VC-bound (3) is actually of the following form (see Cherkassy + Mulier, page 420 for a statement that is more clearly formulated than those in Vapnik's books):

Given a data set $D$ in the unit ball, the set $H_{\Delta,D}$ of all hyperplanes that correctly classify $D$ with margin $\Delta$, see (9.6) on page 419 for a definition of the latter, has VC-dimension $h \leq \min(\Delta^{-2}, N) + 1$. Now, to apply a generalization bound of Vapnik to this VC-estimate one needs to fix this set $H_{\Delta,D}$ of hyperplanes, get a second data set $D_2$, and train a learning algorithm on this second data set $D_2$ that chooses a predictor from $H_{\Delta,D}$. This is the way one needs to interpret the rather informal corollary of Theorem 5.1 in Vapnik's [9] on page 133. Indeed, the same statements can be found in Vapnik's [8] on page 408, where he explicitly refers to results on page 148, in which the class of predictors is, of course, a-priori fixed. (By the way, if the corollary was interpreted to hold on the original data set $D$, instead, then the first term $m/l$, which is the empirical training error, would always vanish.)

Now, if we have a hyperplane $w$ with zero loss on $D$, then it is in $H_{||w||^{-1}, D}$ and one could apply the bounds as described above. But this is far from what is done in the paper.

In summary, the paper has a major technical flaw and for this reason it cannot be accepted.

**Award:**

No

---

### Decision · Program_Chairs · 2022-09-14

Reject